# The Protective and Long-Lasting Effects of Human Milk Oligosaccharides on Cognition in Mammals

**DOI:** 10.3390/nu12113572

**Published:** 2020-11-21

**Authors:** Sylvia Docq, Marcia Spoelder, Wendan Wang, Judith R. Homberg

**Affiliations:** 1Department of Cognitive Neuroscience, Donders Institute for Brain, Cognition and Behaviour, Radboud University Medical Center, 6525 EN Nijmegen, The Netherlands; Sylvia.Docq@radboudumc.nl (S.D.); Marcia.Spoelder-Merkens@radboudumc.nl (M.S.); 2Inner Mongolia Yili Industrial Group, Co., Ltd., Jinshan road 1, Hohhot 010110, China; wangwendan@yili.com

**Keywords:** human milk oligosaccharides, cognition, brain development, animal behaviour, fucosyllactose, sialyllactose, long term potentiation

## Abstract

Over the last few years, research indicated that Human Milk Oligosaccharides (HMOs) may serve to enhance cognition during development. HMOs hereby provide an exciting avenue in the understanding of the molecular mechanisms that contribute to cognitive development. Therefore, this review aims to summarize the reported observations regarding the effects of HMOs on memory and cognition in rats, mice and piglets. Our main findings illustrate that the administration of fucosylated (single or combined with Lacto-N-neoTetraose (LNnT) and other oligosaccharides) and sialylated HMOs results in marked improvements in spatial memory and an accelerated learning rate in operant tasks. Such beneficial effects of HMOs on cognition already become apparent during infancy, especially when the behavioural tasks are cognitively more demanding. When animals age, its effects become increasingly more apparent in simpler tasks as well. Furthermore, the combination of HMOs with other oligosaccharides yields different effects on memory performance as opposed to single HMO administration. In addition, an enhanced hippocampal long-term potentiation (LTP) response both at a young and at a mature age are reported as well. These results point towards the possibility that HMOs administered either in singular or combination forms have long-lasting, beneficial effects on memory and cognition in mammals.

## 1. Introduction

The natural composition of breast milk is well recognized as the golden standard of infant nutrition [1] and is associated with long-term health benefits [2,3,4,5,6,7,8,9,10]. Studies have shown that exclusive breastfeeding is accompanied by a reduced risk for developing medical conditions during childhood such as gastrointestinal infections (e.g., necrotizing enterocolitis) [5,6]. Indications that breastfeeding confers protective effects in the onset and course of allergic diseases such as atopic dermatitis, food allergy and asthma have also emerged over the recent years [7,8,9]. Such protective effects of breastfeeding have been attributed to multiple factors related to the gut, as it is found that breastfeeding can improve immune functioning, promoting a healthy gut microflora [11]. Apart from the gut, bioactive components within breast milk such as the adipokines (e.g., leptin, ghrelin) help regulate appetite control and energy intake. Breast milk also contains growth factors, such as neuronal growth factors (NGF) and epidermal growth factors (EGF), which exert trophic effects on the neonatal nervous system and enhance gastrointestinal mucosal maturation respectively [11,12,13]. In recent years, the mental health benefits that breastfeeding provides have garnered much more attention in neuroscientific research. Notably, breastfeeding is associated with improved cognitive development, as demonstrated by improved IQ scores [14] and a reduced risk of childhood behavioural disorders [15,16]. These findings also coincide with studies showing enhanced brain development parameters, such as white matter development in frontal and temporal regions [17] and maturation of the basal ganglia and thalamus [18]. On the whole, these studies indicate that there are clear developmental and cognitive benefits related to breastfeeding and breast milk, which raises the question: which breast milk factors facilitate cognitive development?

Breast milk is a complex liquid which contains many different lipids (such as the Milk Fat Globule rich in phospholipids and long chain fatty acids), an assortment of vitamins (Vitamin A, B, C, D K), sialic acid (both in free form and bound to oligosaccharides, glycoproteins and glycolipids) and other biologically active components, some of which affect neurodevelopment [19,20,21,22]. Of particular interest to infant nutrition and development are the Human Milk Oligosaccharides (HMOs). These non-digestible carbohydrates are the third most abundant class of breast milk components, and over 200 HMOs, comprised out of 5 monosaccharides (glucose, galactose, N-Acetyl-Glucosamine, fucose and sialic acid) have thus far been identified [23]. HMOs have recently moved into the spotlight of cognitive research due to its widespread effects on infant development and cognition [4,11,20]. There are three main families of HMOs; the non-fucosylated neutral HMOs, (e.g., Lacto-N-neoTetraose (LNnT)), the fucosylated HMOs (e.g., 2′Fucosyllactose (2′-FL)) and the sialylated (SL) HMOs (e.g., 3′Sialyllactose (3-SL) and 6′-Sialyllactose (6-SL)) [23,24]. Oligosaccharides are present in all mammalian milk [25]. However, what makes human milk unique compared to other mammalian milk is that it contains the largest diversity of complex oligosaccharides [25,26] and high concentrations of 2′-FL. It should be noted that the presence of 2′-FL is subject to large inter individual variation depending on the Lewis antigen blood group system of the mother, which encompasses two genes; the Lewis gene (Le gene or FUT-3 gene) and the Secretor gene (Se gene or FUT-2 gene) [27]. Depending on genetic expression, women are either defined as “secretors” (Se+), or “non secretors” (Se-), and Lewis positive (Le+) or Lewis negative (Le-) [23,27]. Both Secretor and Lewis genes are responsible for yielding fucosyltransferase-2 (FUT-2) and fucosyltransferase-3 (FUT-2) respectively, which append fucose to the core oligosaccharides. Depending on which of these FUT enzymes are active, different oligosaccharides will be created; as FUT2 expression results in the synthesis of 2′-FL, while FUT3 expression has been associated with the formation of LNFP-II instead [28,29,30]. These polymorphisms essentially give rise to four major milk groups within the human population, as both genes can be active, inactive or either one of the two is active, hereby resulting in a variable HMO content in breast milk [29]. Around 60–72% of the maternal population are secretors, and the milk of these “secretor mothers” contains an overall higher concentration of HMOs in breastmilk as compared to non-secretors [23,27,31,32]. All in all, a large variability exists within the human population concerning the exact proportions of different HMOs [28]. Moreover, HMOs are also subject to dynamic changes within the same breastfeeding female, depending on factors such as circadian rhythm, lactation stage, maternal diet, and maternal genetic background [4,11,20,28,29,30,31,32,33,34].

Supplementation of infant formula with HMOs renders the composition and downstream effects of infant formula to become closer to those of breastmilk. One of the well-documented advantages of HMOs is its prebiotic role and the capacity to regulate the immune system in the periphery. HMOs can exert antimicrobial and antiviral effects by binding to pathogens which reach the mucosal surfaces in the gut or by directly binding to the gut epithelial receptors, effectively blocking the access of pathogens [11,20]. Experimental studies in infants showed enhancing effects on the immune response of additional 2′-FL supplementation. Goehring and colleagues [35] observed that infants who were fed breastmilk or a 2′-FL enriched formula had lower concentrations of plasma inflammatory cytokines (IL-1α, IL-1ß, IL-6, TNF-α) when compared to children fed the ordinary (non-enriched) infant formula [35]. Furthermore, ex vivo stimulation of peripheral blood mononuclear cells (PBMCs) yielded lower levels of TNF-α and IL-6 when infants were breastfed or were on a 2-‘FL enriched diet. Enriching infant formula with 2′-FL and LNnT also renders the gut microbiome composition and its metabolites (propionate, butyrate and lactate) of formula-fed infants closer to that of breastfed infants [36]. It stands to reason that, if the supplementation of HMOs to infant formula produces immunological and health responses similar to those of breastfed infants, this may also partly account for cognitive outcomes [14]. Indeed, apart from HMOs involvement in immune functioning, a recent study by Berger and colleagues [37] reported that the amount of 2′-FL, measured in mother’s breast milk one month after birth, predicted improved cognitive outcomes in two-year-old children. Since it is known that alterations in the immune system impacts brain development and later life cognitive functioning [38], it is possible that the HMO mediated immune response provides a route via which HMOs could contribute to cognition. Thus, investigating how HMOs impact underlying neural mechanisms of their associated cognitive outcomes will provide valuable insight in HMOs’ role in brain development and functioning.

While there have been correlational studies exploring the role of HMOs on development in humans, no direct human study has thus far investigated both immune and cognitive outcomes with HMO analysis in breast milk or upon HMO supplementation in infant formula. However, direct studies on the effects of HMOs and cognition have been undertaken in murine models and piglets. While there are obvious differences between species, several animal models have been used extensively in behavioural research due to their translational value in brain development and behaviour. The behavioural tasks used in animal models in probing various cognitive functions are well validated [39]. Moreover, since the life span of rodents in particular is relatively short, animal models allow the investigation of the most sensitive developmental period to HMO supplementation. In addition, behavioural studies in animals can be corroborated by more invasive measures in vivo, granting a live view on the underlying neurobiological processes. One method commonly used in rodent memory studies is electrophysiology. Long Term Potentiation (LTP) involves the strengthening of synapses in response to prior stimulation during memory formation and retrieval. This produces a long-lasting shift in synaptic strength and is therefore an important underlying mechanism of synaptic plasticity and memory [40]. Findings derived from preclinical work could prove to be informative and may serve as input to future longitudinal studies on the contribution of HMOs to the cognitive development of humans.

This review’s aim is twofold. Firstly, it aims to summarize the effects of HMOs in animal research and their subsequent cognitive and electrophysiological outcomes. Special consideration is given to the type of HMO used (e.g., fucosylated (2′-FL), neutral (LNnT) and sialylated (3′-SL, 6′-SL)), the age of the animals upon HMO administration, the used cognitive task complexity and the age of the animals during testing. Its second purpose is to provide additional interesting avenues for future research to explore. The search for relevant articles was conducted in Pubmed in the period of 1979 until August 2020, using a specialized search string comprised of both Mesh terms and key words in the title and abstract (Appendix A). This resulted in the inclusion of nine articles that contained (1) an animal model, (2) HMOs and (3) cognitive behavioural tests.

## 2. Assessing the Effects of HMOs on Cognitive Measures in Animal Models

Rodents and piglets are naturally curious and intelligent animals, which results in their frequent use as animal models for the assessment of cognition in a wide variety of behavioural tasks [41,42,43,44,45]. Behavioural tests are considered to be a valid, minimally invasive way to expose underlying cognitive processes, under the condition that the animal is capable of, and facilitated in, expressing such processes externally. In the context of HMO research, the focus has mainly been on memory and learning behaviour as cognitive capabilities. In the following sections, we will first graphically present an overview of the animal tests which investigated the consequences of HMOs on cognition. Subsequently, we present the main findings of the selected nine articles, grouped by the type of HMO (fucosylated or sialylated), in Table 1. Thereafter, the main results will be described, which is then followed by a discussion about the implications of the findings reported in the investigations.

The type of behavioural tests used to study the effects of HMOs on cognition make use of either the intrinsic rewarding value of an animal’s natural curiosity in new exposures (Figure 1A,B,E) [41,42], the aversion to uncontrolled swimming without a platform to rest on (Figure 1D) [43] or the willingness to obtain an extrinsic reward like food or water (Figure 1C,F,G) [44,45]. Since animals prefer to be exposed to new items or environments to explore, the time spent to explore this new item or environment can be used as a measure for spatial or recognition memory. The willingness to obtain a food or water reward is commonly measured in operant conditioning tasks in either a skinner box or an Intellicage [44,45]. Operant conditioning tasks encompass associative learning paradigms, in which certain behaviour is reinforced via a reward or a punishment. In operant conditioning, different reinforcement schedules exist, such as the Fixed Ratio (FR) schedule, in which animals have to reach a certain criterion before they receive a reward. For example, an FR(4) schedule requires 4 correct responses from the animal in order for it to obtain a reward.

Overall, these cognitive tasks can be grouped by the level of complexity, as tasks that require a few trials are considered to be easier to perform than a task that requires weeks of training. In light of this, we have grouped the Y maze, T maze, Morris Water Maze (MWM) and the Novel Object Recognition Test (NORT) as simple cognitive tests and the 8-arm radial maze and the operant tasks (Skinner box and Intellicage) as the complex cognitive tasks.

## 3. Effects of HMOs on Cognition in Mammals

### 3.1. Main Behavioural Findings

Supplementing mammals with additional HMOs leads to beneficial cognitive outcomes under certain specific circumstances (Table 1, Figure 2). In general, both fucosylated and sialylated HMOs contribute to an improved memory performance and faster learning speed (tests described in Figure 1A–G) when tested in mature adulthood, irrespective of the age of administration of these HMOs (e.g., during infancy or adulthood) [46,47,48,49,50,51,52,53,54].

#### 3.1.1. Simple Cognitive Tasks

When rodents performed spatial and recognition memory tests during adolescence and early adulthood, no effects of either fucosylated or sialylated HMOs, as assessed by the NORT (when tested 24 h after the acquisition phase), MWM and the Y maze, were reported. Contrary to the rodent studies, three piglet studies showed that supplementing HMOs during the lactation period resulted in improved spatial memory (T maze) in infancy [51] and object recognition (NORT) [52,53]. Supplementing only oligofructose or the combination of 2′FL and LNnT increased object recognition when piglets were tested one hour after the acquisition phase. When tested 48 h later, only the piglets who received a combination of either Bovine Milk Oligosaccharides (mostly neutral non fucosylated oligosaccharides) and 2′FL and LNnT [52] or Oligofructose and 2′-FL [53] displayed long-term recognition memory. In mature adulthood (older than 1 year), rodent studies also found significant differences in both the Y maze and the NORT for both sialylated and fucosylated HMOs. However, the sialyllactose piglet study performed by Fleming and colleagues [54] yielded no results. In this study, they found no differences between the sialyllactose group and control group on the NORT performed during infancy.

#### 3.1.2. Complex Cognitive Tasks

When considering the tasks that probe conditioning and learning capabilities and in which the cognitive difficulty could be varied, such as the 8-arm radial maze [50] and operant tasks [47,48,49], beneficial effects of HMO already surface at a young age in rats, mice and piglets alike. These effects also persist throughout adulthood. Perhaps the beneficial effects of HMOs become especially apparent upon increments on the cognitive load to meet the task demands.

### 3.2. Effects of HMOs on Long Term Potentiation (LTP)

The method of in vivo LTP induction in the studies listed here involved the implanting of stimulating electrodes on the Schaffer’s collateral of the dorsal hippocampus and 2 to 4 recording electrodes in the stratum radiatum underneath CA1 [46,47,48,49]. A high frequency stimulation (200-Hz trains of pulses, 100 ms each and presented repeatedly with 1-min intervals) was delivered to the Schaffer’s collateral and 30 min later the field excitatory post-synaptic potentials (fEPSPs) were recorded. Enhanced LTP responses are reported in all these studies [46,47,48,49], both after weaning and during adulthood, when animals were supplemented with fucosylated or sialylated HMOs.

## 4. Discussion

To the best of our knowledge, this review is the first to summarise the effects of fucosylated and sialylated HMOs on cognition and electrophysiological brain recordings in rodents and piglets. The effects of both types of HMOs uncovered in the reported investigations unequivocally point towards long lasting beneficial effects on cognition and memory, which is further supported by changes in the underlying physiological mechanisms as measured by LTP [46,47,48,49].

The majority of the reported animal studies, included in Table 1, revealed that HMOs enhance learning and memory. For the simple cognitive tasks, the effects of HMOs are not unequivocal, as differences are observed between the animal model used, task parameters, the dosage used and age of administration and testing. It should be noted that in the majority of the studies, the HMO dosage was comparable to concentrations found in human milk [28,29,30,31,33], and effects of HMO supplementation were already visible at these physiological relevant dosages.

In rodents, no significant effects on spatial memory or long-term recognition memory are reported when the animals’ age ranges from juvenile to young adulthood. In piglets, HMOs are found to affect spatial memory and intermediate recognition memory but not long-term recognition memory when they were fed only HMOs during infancy. Inter species differences between rodents and piglets may help to explain why effects of HMO administration are visible in piglets but not in rodents when tested at a very young age. The third trimester in human gestation coincides with the first ten postnatal days of rat pups, while the neurodevelopmental trajectory and morphological properties of piglet brains are much more comparable to humans [41,55,56,57]. This complicates the comparison of the effects of oral delivery of HMOs between piglets and rodents. Differences reside in the immediate environment upon birth and the extent to which the brain and body have developed at that point, as neonatal rat pups would be more comparable to prenatal piglets in the final days before parturition, and there are no studies performed on the cognitive effects of HMOs on piglets in young adulthood. This interspecies difference in developmental stage upon birth and subsequent postnatal period might contribute to the heterogeneity in the findings between species on simple behavioural tests such as the NORT and the T maze.

Nevertheless, one cannot exclude the possibility that other factors than mere species differences may be at play, for example, the test parameters used in the studies. In the NORT of the rodent studies, the retention interval (time between acquisition phase and test phase) was 24 h, which is considered to be fairly long and is considered to be a measure of long-term recognition memory [42]. In the piglet studies, different retention intervals, ranging from 1 h (intermediate) to 48 h (long-term), were used. It is possible that similar enhancing effects of HMO administration on recognition memory (NORT) reported by Fleming and colleagues [52,53] would have been found in juvenile rodents if the retention interval was 1 h instead of 24 h and if the rodents had been fed a similar combination of oligosaccharides as the piglets received. However, when probing such a long-term recognition memory of one-year old rodents, an improved recognition memory is observed in the HMO supplemented animals, together with improved spatial memory as measured by the Y maze. As long-term recognition memory was not observed in juvenile piglets and rodents when supplied with only one HMO but was observed in piglets when they were given a combination of oligosaccharides, this may not be a simple matter of species differences. Another explanation could be that within the developing brain, there are different processes at play when retrieving a newly consolidated memory (one hour later) versus an older memory (24–48 h later), which may require more resources, such as the combination of various types of oligosaccharides. Interestingly, when piglets were supplemented with a complex mixture of oligosaccharides (HMOs and BMOs or Oligofructose), they displayed an improved long-term recognition memory. Perhaps the effects when HMOs form combinations or are provided with other oligosaccharides are more potent and thus easier to discern than the effects of singular HMOs on memory.

Other factors such as gender and sample size could also contribute to the heterogeneity of the simple behavioural test findings, but it is uncertain to what extent these factors may have influenced the results. Only two sialyllactose studies (and no fucosyllactose study) used both males and females, one rodent study by Oliveros and colleagues [47] and one piglet study by Obelitz-Ryom and colleagues [51]. However, no separation based on gender was performed in the analysis. As studies on postnatal administration of compounds, such as the study by Shumake and colleagues [58] have demonstrated gender specific effects in rats, it stands to reason that early life HMO supplementation could produce gender specific outcomes. Nonetheless, when comparing the findings generated by Oliveros et al. [47] and Obelitz-Ryom et al. [51] with the exclusively male studies of the same species and HMO administered, the behavioural results remained very similar. Furthermore, the majority of the studies employed comparable sample sizes (*n* = 10–12 on average), and effects of HMOs on cognition were already reported in studies with the lower sample sizes. While potential effects of variation in sample size cannot be completely excluded, HMO supplementation already produces beneficial results in experiments with lower sample sizes. Therefore, the heterogeneity in findings between studies is more likely due to a combination of factors such as species and task parameters, as previously discussed.

When both piglets and rodents were tested on complex cognitive tasks from a young age onwards, HMOs exerted a beneficial effect on learning and memory. Therefore, it is possible that the HMOs effects become more apparent when cognitive load is increased, either due to task difficulty or due to aging. This may explain why the beneficial effects of HMOs are especially visible when the tasks are cognitively more strenuous, such is the case with the 8-arm radial maze or the operant tests, as increases in cognitive load make brain limitations more discernible.

While behavioural tests on learning and memory at a young age in general yielded mixed results, HMO supplementation did significantly improve LTP from a young age onwards. Interestingly, while in both young adult (2.5 months old) and mature adult (1 year old) just one HFS application was sufficient to induce LTP, very young rodents (6 weeks old) required a second high-frequency stimulation (HFS) to induce LTP. Nonetheless, HMO administration resulted in an enhanced LTP response in both younger and older rodents alike. It is possible that LTP might be a more sensitive measure to investigate the beneficial effects of HMOs on cognitive outcomes at a young age. Furthermore, under normal circumstances, the LTP response is reduced in older rats as a natural result of aging [46]. This natural reduction in LTP response was not encountered when the animals were supplemented with HMOs. On the contrary, supplementation with HMOs facilitated an enhanced LTP response. Because LTP is a measure of synaptic plasticity, it stands to reason that synaptic plasticity benefits from HMOs both in the short-term as in the long-term. Therefore, supplementation of HMOs, both sialylated and fucosylated, in infancy could have long-lasting protective effects on the molecular underpinnings of learning and memory.

It should be noted that these results have been gathered from only nine articles, which is the main limitation of the present review. Nevertheless, while there are a limited number of studies on the cognitive effects of HMO supplementation, the studies currently available show promising results of how HMOs could contribute to cognitive development. These findings call for further in-depth research on the cognitive effects of HMOs and to delineate their underlying mechanisms.

### Potential Underlying Mechanisms

There are a few possible factors which could account for the cognition enhancing effects of HMOs in mammals.

In the case of sialylated HMOs, Polysialylated Neural Cell Adhesion Molecules (PSA-NCAM) could be upregulated. The PSA-NCAM complex is upregulated in newborn, immature neurons and growing fibre tracts during embryogenesis and has been linked to increased synaptic plasticity [59,60,61,62]. Within the adult brain, PSA-NCAM is expressed in brain regions with high rates of neural plasticity and neurogenesis, such as the olfactory bulb and the hippocampus [61]. Improved neural plasticity and the survival of newborn neurons contribute to cognition and memory [62]. Therefore, it is possible that sialylated HMOs are capable of influencing neurogenesis via upregulation of PSA-NCAM, which in turn contributes to the reported improvement in cognition. This suggestion is further supported by Oliveros and colleagues [47]. These authors found an increase in PSA-NCAM in 6-SL supplemented animals. However, the role of fucosylated HMOs in plasticity and neurogenesis is currently not well understood and requires further investigation.

A second possible factor is the improved immune functioning due to the supplementation of HMOs and their well-established role in the immune system. As mentioned in the introduction, immune factors also contribute to cognitive functioning [38], though there are multiple hypotheses on how this may occur. One hypothesis states that perinatal immune activation directly affects neurodevelopmental pathways necessary for learning and memory, which leads to reduced neurotransmitter function, a reduction in hippocampal presynaptic proteins and impaired LTP [38]. A second hypothesis postulates that early life immune activation indirectly determines the adult response to an infection with a pathogen, either via exaggerated pro-inflammatory cytokines or via a decrease in anti-inflammatory cytokines. This in turn could lead to downstream changes in cognition and neural function [38]. As HMOs are capable of regulating the neonatal cytokine response in the periphery [11,33,35,63], it is possible that they also exert their enhancing effects on cognition via the immune system.

A last possible factor through which HMOs may improve cognition involves the microbiome. HMOs contribute to the microbiome composition within the gut and therefore could interact with the brain via the resulting bacterial metabolites such as the Short Chain Fatty Acids [64]. As certain gut bacteria are specific for the utilization of sialylated HMOs and other bacteria for the fucosylated HMOs, a larger variety of HMOs may go hand in hand with a larger yield of specific gut bacteria capable of metabolizing these HMOs, and thus determining their subsequent metabolites [65]. Interactions between single HMOs and the microbiome have been previously reported by Tarr and colleagues [66]. They demonstrated that the administration of sialylated HMOs changed the microbial composition in the gut of mice, which in turn led to a reduction in anxiety-related behaviour and a maintenance of neurogenesis. The influence of the gut–brain axis has also been touched upon by Vazquez and colleagues [48], as they found that ablating the vagal nerve, which is part of the gut–brain axis, diminished the beneficial effects of orally supplied 2′-FL on LTP. Similar to these results, Kuntz and colleagues examined the metabolic fate of 2′-FL and found that 2′-FL was not directly incorporated in the brain but required an intact gut microbiome for the generation of fucose metabolites, which are subsequently taken up into the systemic circulation and organs [67]. In addition, it is possible that combinational HMOs may generate better effects than alone. This idea has already been demonstrated at the level of the growth and function of gut bacteria [68,69]. Different HMOs are processed by different bacteria, which contain either sialidases or fucosidases to cleave sia and fuc of the carbohydrates [65]. In turn, another group of bacteria can feed on the HMOs once the fuc and sia moieties are removed. These bacterial interactions, which depend on the HMOs present in the gut, may exert downstream effects on memory and cognition via the gut–brain axis. In light of potential downstream effects of the microbiome on behaviour, environmental housing conditions which affect the microbiome should also be considered [70] in this context, although it is uncertain to what extent the microbiotic variations due to husbandry may have influenced the effects of HMO supplementation on subsequent behaviour. Finally, another important factor to consider in the context of the microbiome is gender specific effects. While infant sex is reported to be largely unrelated to the HMO composition within human breastmilk [31], another study by Moossavi and colleagues [71] found that the milk microbiota vary depending on infant sex. This could potentially be attributed to cross interactions with the gut microbiome of the infant, as gender differences have been reported there [71]. As HMOs interact with both the milk and the gut microbiome [72,73], it is therefore possible that sex-dependent variations could lead to differential cognitive outcomes of HMO supplementation.

## 5. Conclusions

The observation that HMOs are capable of enhancing cognition has initiated the search for a better mechanistic understanding of its functioning. Nonetheless, there are still several outstanding questions on the relationship between HMO and neonatal brain development, which warrant further investigation. An important aspect that needs to be addressed is the apparent age-related differences when assessing various cognitive tests. This point illustrates one of the current issues on HMO research in animals, as the tools currently used may not be sensitive enough to fully explore the range to which HMOs may affect brain development and cognition. Thus, one of the more complex tools could be the use of challenging operant tasks, such as the Trial Unique Delayed Non-Matching to Location (TUNL) measuring spatial memory and pattern separation [74], the 5-Choice Serial Reaction Time Task measuring attention and motor impulsivity [75], or delayed reinforcement tasks measuring choice impulsivity [76], ideally performed in the animal’s home cage. The difficulty of such tasks can be varied and may thus be more suited to test cognitive functioning in young animals, as at a young age, only effects of HMOs were found in difficult tasks.

Another important issue is that due to the large variability between the experimental design and methods used across studies, comparing the effects of different HMOs between studies is difficult. Such variability includes the age of testing, the tests and experimental parameters, the HMO components used, the gender of the animals, variation in sample sizes, the environmental conditions and the variation in (neuro)developmental stage during which the animals were supplemented the HMOs. These limitations call for a larger, unified study in which the effects of different HMOs on complex cognitive functioning are systematically compared, when administered both independently and as in conjunction. In such a unified study, all these factors can be accounted for, enabling a systematic comparison.

A last important issue is that most HMO studies so far have focused on singular HMOs, with the exception of the two most recent studies performed by Fleming in 2020. The focus on singular HMOs is a limitation because it does not reflect a naturalistic situation where maternal milk provides a combination of different HMOs [77]. Therefore, considering the interactions of HMOs when supplemented in combination would provide valuable insights on the influence of the gut microbiome and its downstream effects on cognition and development.

While research on the cognitive implications of HMOs is still in its infancy, the early findings reporting its long-lasting beneficial effects on memory and cognition are promising. Further studies on the exact molecular mechanisms, ranging from immune functioning to neuroplasticity and the microbiome will prove to be useful in deepening our understanding of how HMOs and their interactions contribute to cognition and development.

## Figures and Tables

**Figure 1 nutrients-12-03572-f001:**
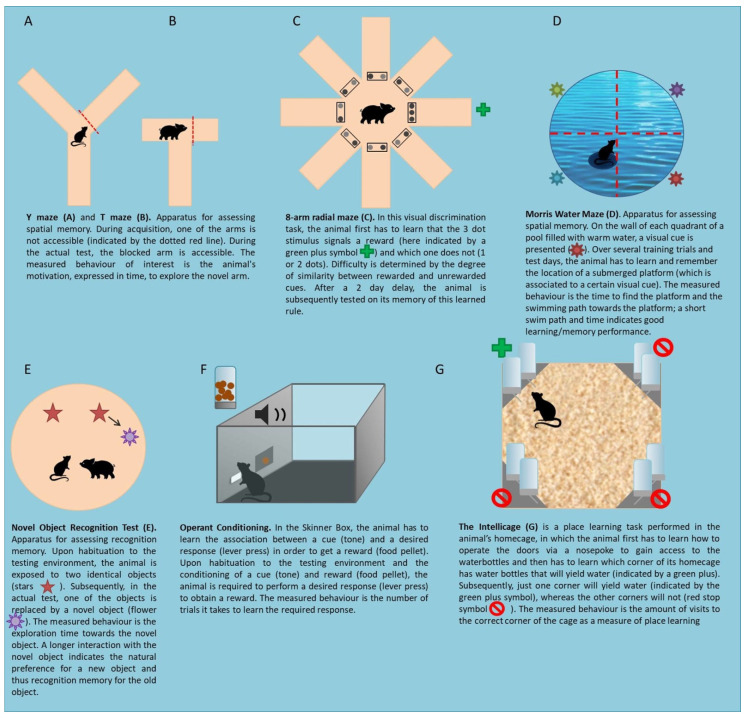
Summary of the behavioural tests used in the HMO studies. The type of animal placed inside the test (rodent or piglet) corresponds to the animal model used in the behavioural paradigms included in this review.

**Figure 2 nutrients-12-03572-f002:**
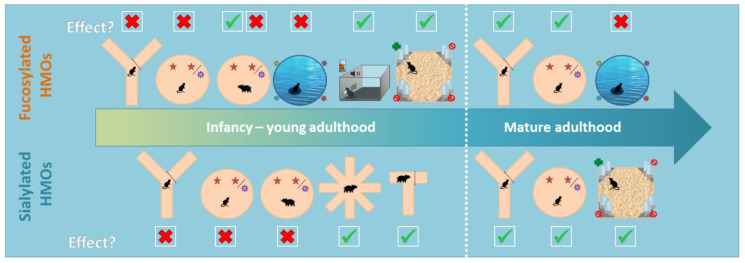
Graphical summary of behavioural tests results. The results have been grouped based on the type of HMO (Fucosylated versus Sialylated), animal model (rodents versus piglets) and the age of when the behavioural test has been performed. Infancy–young adulthood has been defined as the period ranging from PND1–6 months of age, while mature adulthood encompasses animals of 1 year old. Red crosses indicate that no significant differences were observed between the HMO and the control group, while green check marks indicate that positive effects due to HMO supplementation were reported. Details on the nature of such effects are summarized in Table 1.

**Table 1 nutrients-12-03572-t001:** Summary behavioural studies.

Study	Species	HMO Component and Dose	Age and Duration Administration	Age Test	Tests	Key Results
**Fucosylated HMOs**						
Oliveros et al., 2016 [46]	Lister Hooded Rats	2′-FL (1g/KG/BW)	From PND 3–weaning	Long Term study: (1) 4–6 weeks (2) 1 year old	NORT Y maze MWM LTP (only at 1 year)	At 6 weeks of age no differences in behaviour (*n* = 12) were found. At 1 year of age, 2′-FL rats showed improved performance in the NORT and Y- maze paradigms. No effect was observed in the MWM. LTP was more intense and long lasting in the 2′-FL supplemented groups (*n* = 10)
	Sprague Dawley Rats	2′-FL (1g/ KG/BW)	From PND3 until week 6	Short Term study: 6 weeks	LTP	LTP was more intense and long lasting in the 2′-FL supplemented groups (*n* = 10).
Vazquez et al., 2016 [48]	Sprague Dawley Rats	2′FL (350 mg/kg/BW via AIN-93M diet) L-Fucose (Fuc) (equimolar amounts of fuc and 2′-FL via AIN-93M diet)	3–4.5 months old for 5 weeks	Started at 2.5–4 months old	Operant conditioning (FR1) LTP	2′-FL but not fuc displayed enhanced LTP. Vagotomy inhibited the effects of oral 2′-FL on LTP (*n* = 10) and operant learning paradigms (*n* = 10).
Fleming et al., 2020a [52]	Pigs (1050 Cambro genetics)	Three groups: Oligofructose (OF) 5 g/L OF + 2′-FL 5 g/L OF + 1 g/L 2′-FL Control. Nothing	PND 2–33	PND 22	NORT	Pigs (*n* = 12) who received Oligofructose (OF) displayed enhanced object recognition when tested 1 h after being habituated to the two objects. When pigs consumed both 2′-FL and OF, they showed improved recognition memory after a 48-h delay.
Fleming et al., 2020b [53]	Pigs (1050 Cambro genetics)	Four groups: HMOs (2′FL + LNnT) 1 g/L 2′-FL + 0.5 g/L LNnT BMOs 12.4 g/L HMOs + BMOs 1 g/L 2′-FL + 0.5 g/L LNnT + 12.4 g/L BMOs Control. Nothing	PND 2–33	PND 22	NORT	Pigs (*n* = 12) who received only HMOs displayed enhanced object recognition when tested 1 h after being habituated to the two objects. When pigs consumed both HMOs and BMOs, they showed improved recognition memory after a 48-h delay.
Vazquez et al., 2015 [46]	Sprague Dawley Rats	2′FL (1 g/kg/BW) via oral gavage during acute administration and 2′-FL (350 mg/Kg/BW) via AIN-93G diet, during short time feeding	Acute administration: when rats were 3 months old Short-time feeding from 2.5–4 months, for 5 weeks	Operant tests started when administration started. LTP was performed after administration period.	Operant conditioning (FR1) LTP	2′-FL groups performed better in operant learning paradigms (rats *n* = 10, mice *n* = 28) and showed an enhanced LTP response (rats and mice *n* = 8). The long-time supplementation of 2′FL also increased the expression of molecules involved in storage newly acquired memories (BDNF, PSD-95 phosphorylated CamKII, etc.)
	C57BL/6 mice	2′FL (350 mg/Kg/BW via AIN-93G diet)	Long-time feeding from 2–3.5 months old, for 12 weeks		Intellicage (FR1, FR4, FR8) LTP	
**Sialylated HMOs**						
Oliveros et al., 2018 [47]	Sprague Dawley Rats	Neu5Ac 6′-SL (Dose ranged from 400 mg/Kg/BW to 2600 mg/Kg/BW based on theoretical model)	From PND 3 until weaning	After weaning	NORT Y maze	No effects detected after weaning (*n = 10*). At 1 year old, sia (Neu5Ac and 6′-SL) exposed rats (*n* = 8*♀*) showed improved performance on all the behavioural tests (NORT, Y-maze, Intellicage) and showed enhanced LTP (*n* = 10) when compared to the control group. Of the SL supplemented animals, the 6′-SL group performed better than the Neu5AC group
				1 year old	NORT Y maze Intellicage LTP	
Wang et al., 2007 [50]	Piglets Landrace/Large White cross	Sialic Acid (ingredient of Casein glycomacropeptide cGMP)) (4 groups of animals with their own dose each; 0 mg/L (control), 140 mg/L; 300 mg/L; 635 mg/L and 830 mg/L)	From PND 3 until end of experiment	PND 21–PND 35	8-arm Radial maze	Supplemented groups *(n =* 12–14 *per group)* required less trials to learn the required response, with a dose–response correlation for the difficult task.
Obelitz-Ryom et al., 2019 [51]	Pre-term delivered (experimental groups) Piglets *Landrace x Yorkshire x Duroc*	Sialyllactose (6′-SL + 3′-SL) (380 mg/L) Lactose (control) (6000 mg/L)	PND1–PND19	PND13–PND18	Spatial T-maze	Four experimental groups were included in the study; PRE-SAL (*n* = 10 *♀*, 10 *♂*), PRE-CON (*n* = 9 *♀*, 11 *♂*), TERM-CON (*n* = 9 *♀*, 5 *♂*) and TERM-SAL (*n* = 6 *♀*, 6 *♂*). TERM CON piglets reached learning criteria of 80% correct choices on day 3, PRE-SAL on day 4 and PRE-CON on day 5. More PRE-SAL piglets reached the T maze learning criteria compared to PRE-CON piglets. Upregulation of genes for sialic acid metabolism, myelination and ganglioside biosynthesis were present in the hippocampus of SL supplemented preterm piglets.
	Term delivered piglets (reference groups) *Landrace x Yorkshire x Duroc*	Lactose (control) (6000 mg/L) Pig’s milk (under natural rearing conditions)				
Fleming et al., 2018 [54]	Piglets (no breed specified)	Sialyllactose (380 mg/L)	PND2–PND22	PND15–PND22	NORT	No effects (*n* = 17) were observed.

NORT: Novel Object Recognition Test, MWM: Morris Water Maze, LTP: Long-Term Potentiation. BMO: Bovine Milk Oligosaccharide. When provided, strains of species have been included in the table. In all studies presented here, the HMOs were administered orally. All animals used in the studies were male, unless otherwise specified. When the experimental groups have not been detailed in the key results column, the reported n indicates the number of animals per experimental group of that study.

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
