# Peer review of "The Protective and Long-Lasting Effects of Human Milk Oligosaccharides on Cognition in Mammals"

_nutrients, 2020, doi:10.3390/nu12113572_

Round 1

Reviewer 1 Report

Overall comments

The paper by Docq et al. is a review of HMO and their effects on memory and cognition in mammals. The review is based on nine published articles in Pubmed since 1979.

The current review excels in some aspects: the literature search strategy was appropriate and transparent, the coverage of the literature comprehensive, and the review well organized. In addition, the authors describe the limitations to the current literature, and also provide statements which synthesize the evidence and limitations into clear “take home messages” representing the overall conclusions that can be drawn. However, the review remains weak, as only a limited number of papers can be included.

My major concern is related to the overall impact taken into consideration that only nine papers exists on the topic. Thus, I would question the usefulness of a review limited to nine papers. Readers embarking on studies in this field could easily grasp those nine papers. Further, it is mentioned that out of these nine papers, seven are selected for an in depth look (Line 116).

Minor comments

In the manuscript milk oligosaccharides are generally abbreviated HMO, which implies human origin. Infant formula supplemented with synthetically produced 2FL does not contain HMO, since it is not from human origin. Thus, I suggest to avoid using the abbreviation HMO for oligosaccharides and using OS or MO instead, unless the OS/MO are of human nature, then HMO are applicable.

Line 46, Sialic acid is mentioned. Is this free-form or a part of HMO?

Line 48, What is the meaning of non-soluble HMO? I suppose all components of human milk are present in the liquid solution?

Line 53, These abbreviations (LNnT, 2FL, 3SL, 6SL) have not been defined.

Line 55. The sentence suggest that Human OS are present in milk from other mammals, which of course is not true. In bovine milk, the  oligosaccharides are generally termed BMO. I suggest to rephrase the sentence.

Line 57, I think it will be valuable to include an introduction to secretor and lewis systems, as these do have a general impact on HMO profile.

In section ”2. Assessing the effects of HMOs on cognitive measures in animal models”, besides Table 1, there is not a single reference to existing literature in this section. In my opinion it is not enough to reference one table and then write the main text without further references.

Author Response

Dear reviewer,

Thank you for your time and your feedback, we found your comments very helpful. We have responded to your comments in blue below and implemented the changes in the manuscript in red. 

Response to Reviewer 1 Comments

Point 1: My major concern is related to the overall impact taken into consideration that only nine papers exists on the topic. Thus, I would question the usefulness of a review limited to nine papers. Readers embarking on studies in this field could easily grasp those nine papers. Further, it is mentioned that out of these nine papers, seven are selected for an in-depth look (Line 116).

REPLY: We understand your point of concern. However, we have several reasons for our choice to include a specific selection of papers and why we believe the publication of this focused review is timely. We have set out these arguments by a few separate points we would like to address:

The search string we used in Pubmed, as described in Appendix 1, provided us with 108 articles at the time. This search string enables the detection of articles that contain information regarding HMOs, their cognitive outcomes, and specific HMO components. However, the main body of the initial 108 papers focused on the role of HMO’s on immune functioning and disease, the gut microbiome and breast milk. By further focusing on the detailed topic we wished to address, namely, experimental studies in which animals were exposed to both the administration of HMO’s and the measurement of cognitive parameters, our literature search finally resulted in 9 articles.

  1. The reason why we decided to write a review on this focused topic is because it was unclear for us, based on the available literature,
    • which type of administrated HMO (fucosylated, sialylated, or other oligosaccharides such as Lacto-N-Tetraose))
    • when during development,
    • and with which frequency and duration of administration,

cognition was altered, and in which cognitive domain such alterations occurred. We were most interested in memory and learning. Therefore, we slightly disagree with the reviewers’ statement that ‘Readers embarking on studies in this field could easily grasp those nine papers’. Due to the experimental differences in those papers and heterogeneous results, readers are unable to quickly draw a main conclusion. Therefore, this review helps to guide readers/researchers to when and which HMOs have significant effects on learning and memory.

  1. Another reason why we decided to write a review on this topic is that we did not have the sole intention of merely summarizing the existing literature out there, but it was also our aim to critically reflect on the studies that have been performed and provide additional interesting avenues for future research to explore. In this sense, the fact that there are only nine papers on this particular subject, while there are still so many questions on the effects of HMOs on cognitive development, only further emphasizes the need for more research, preferably performed in such a way that the results can be more readily compared. We think that calling attention to this and providing suggestions for future research, based on the current findings and limitations, is also scientifically valuable.
  2. Furthermore, we believe that, if we expand the review and add more articles and thus broaden the scope of the review, we lose focus of the paper.

Having said this, we realize that the purpose of the review has not been explained in such detail in the manuscript. We have clarified and expanded the purpose of the review on page 3 line 122-127. We hope that in doing so, the relevance of this review has been more accentuated. 

Regarding line 116, thank you for pointing it out, it seems we have forgotten to adjust the number during our final edits. We have corrected it in the text now and adjusted it from ‘seven’ to ‘nine’, please see line 139 on page 3.

Point 2: In the manuscript milk oligosaccharides are generally abbreviated HMO, which implies human origin. Infant formula supplemented with synthetically produced 2FL does not contain HMO, since it is not from human origin. Thus, I suggest to avoid using the abbreviation HMO for oligosaccharides and using OS or MO instead, unless the OS/MO are of human nature, then HMO are applicable.

REPLY: Thank you for your suggestion, we understand your point. While the 2-FL ingredient which has been added to infant formula, is not of direct human origin (including other supplemented HMOs such as LNnT, 3-SL and 6-SL), it is considered to be equivalent to HMOs obtained from reference material isolated from human milk, please see also the following FDA approved oligosaccharide documentations such as https://www.fda.gov/media/124475/download and https://www.fda.gov/media/100020/download; and also an example of the safety evaluation report from EFSA https://www.efsa.europa.eu/en/efsajournal/pub/4184. These synthetically produced oligosaccharides are also commonly referred to as Human Milk Oligosaccharides in other papers (see also https://www.ncbi.nlm.nih.gov/pmc/articles/PMC6213476/) including the ones we discuss, because most of these oligosaccharides (such as 2’-FL), are not found in animal milk of other species, but only appear in human milk. This is why we have referred to these oligosaccharides as ‘HMOs’ and in order. 

Point 3: Line 46, Sialic acid is mentioned. Is this free-form or a part of HMO?

REPLY: Sialic acid in breastmilk is mainly present in a bound form as a functional group inside oligosaccharides, glycoproteins and glycolipids but can also appear in free-form. We have now specified these sialic acid properties in line 52.

Point 4: Line 48, What is the meaning of non-soluble HMO? I suppose all components of human milk are present in the liquid solution?

REPLY: We meant to indicate that these are not digested by the enzymes in the upper gastrointestinal tract, though in hindsight we should have called them ‘non-digestible’ to avoid confusion.
We have amended the phrasing to “non-digestible carbohydrates” on page 2, line 55.

Point 5: Line 53, These abbreviations (LNnT, 2FL, 3SL, 6SL) have not been defined.

REPLY:  Thank you for pointing this out. We have addressed these abbreviations and included the full name of the components. Please see page 2, line 60-61.

Point 6: Line 55. The sentence suggest that Human OS are present in milk from other mammals, which of course is not true. In bovine milk, the  oligosaccharides are generally termed BMO. I suggest to rephrase the sentence.

REPLY: We have addressed this point and changed the sentence to “Oligosaccharides are present in the milk of other animal species” in line 62.

Point 7: Line 57, I think it will be valuable to include an introduction to secretor and lewis systems, as these do have a general impact on HMO profile.

REPLY: Thank you for your suggestion. We have included an introduction on the secretor and Lewis systems and how they affect HMO profiles in breast milk, please see lines 64 – 77.

Point 8: In section ”2. Assessing the effects of HMOs on cognitive measures in animal models”, besides Table 1, there is not a single reference to existing literature in this section. In my opinion it is not enough to reference one table and then write the main text without further references.

REPLY: We agree with your point and have included appropriate references for the behavioural tests we describe in that section, we refer to page 3 (line 134) and page 4 (lines 149, 150, 154) for the additional references. The reference list has been updated to reflect these additions as well.

Reviewer 2 Report

Docq et al present a review examining the effects of HMOs on cognition in mammals.

Abstract:

The abstract is fine but the authors might consider making it a little less vague. Perhaps state the memory and learning outcomes that are specifically dealt with in the review? Also, are there any particular species of mammal that the work focuses on?

Introduction:

Well thought out and thorough introduction to the literature in relation to HMOs and neurocognitive and immunological outcomes. Well done! I have 2 very minor points,

Line 32 – celiac disease is an inherited condition, perhaps the risk of developing severe symptoms may be reduced but the risk of developing celiac disease would not be reduced. The authors should reword this statement.

Line 35 – While this is all true and very valid the authors should also make clear that the positive effects on the gut are not the only way in which breastfeeding elicits its positive health effects (adipokines and growth factors help establish normal satiety responses etc). A brief statement should be included for clarity.

Main body of the review:

The authors have put together a comprehensive overview of the literature in this area using defined search terms and a systematic approach. I’m therefore curious to know why the authors didn’t take it a step further and develop a systematic review? The article clearly shows there is research in the area warranting such an approach.

As the authors have used the more systematic approach it disrupts the overall flow of the manuscript. It might have been clearer and more in line with a traditional narrative review to incorporate the behavioral findings and the mechanisms in the same sections to allow the reader to tie the two areas together with greater ease?

Figure 1 and 2 gives a really nice visual overview of the types of behavioral tests and overall results. A really nice addition.

Could the authors comment on the physiological relevance of the doses used in these studies. Would there be grams of each HMO in actual breastmilk or the human infant formula that contains HMOs?

Rodents are altricial and pigs are precocial, how does this affect brain development and might this account for some of the differences between the species in this line of research?

In the discussion please make sure to identify which species the cited studies are referring to. This is especially important given that there are differences in the effects of HMO supplementation between piglets and rodents in the outlined studies.

The authors point out the contribution of the gut and in particular the microbiome in mediating the effects of HMOs. Would the environments that each species are housed in influence the microbiome of each species in a way which may dictate the overall results?

Each of the species studied produces litters, is there any information in relation to the sex of the animals who underwent the behavioral tests and might this influence the results? There is a growing body of result to suggest quite significant sex-specific outcomes in a range of different organ systems in response to supplementation (particularly when it is administered early in life) so a comment on sex-specificity and whether this may influence the ability of HMOs to produce an impact of neurocognition should be commented on. If you can find any information on whether human mothers produce differing amounts of specific HMOs in response to infant sex that would also be worth mentioning.

Line 284-285 – This sentence is confusing and should be reworded.

Author Response

Dear reviewer,

Thank you for your time and your feedback, we find your comments very helpful. We have replied to your comments in blue below and the changes in the manuscript are indicated in red. 

Response to Reviewer 2 Comments

Abstract:

Point 1: The abstract is fine but the authors might consider making it a little less vague. Perhaps state the memory and learning outcomes that are specifically dealt with in the review? Also, are there any particular species of mammal that the work focuses on?

REPLY: Thank you for your comments. We have amended the abstract to specify the findings discussed in the review and we also included the specification of the species (rats, mice and piglets) in the abstract to clarify which mammals were included in the studies, please see lines 14 – 23 on page 1.

Introduction:

Well thought out and thorough introduction to the literature in relation to HMOs and neurocognitive and immunological outcomes. Well done! I have 2 very minor points,

Point 2: Line 32 – celiac disease is an inherited condition, perhaps the risk of developing severe symptoms may be reduced but the risk of developing celiac disease would not be reduced. The authors should reword this statement.

REPLY: Thank you for the compliment and for pointing out the issue with this statement, we have looked up the most recent literature and based on the latest information which does not provide unequivocal support for the benefits of breastfeeding and celiac disease, we have decided to omit the mentioning of celiac disease from the introduction (line 33).

Point 3: Line 35 – While this is all true and very valid the authors should also make clear that the positive effects on the gut are not the only way in which breastfeeding elicits its positive health effects (adipokines and growth factors help establish normal satiety responses etc). A brief statement should be included for clarity.

REPLY: We have included the role of adipokines and growth factors in the introduction, please line 37-41.

Point 4: The authors have put together a comprehensive overview of the literature in this area using defined search terms and a systematic approach. I’m therefore curious to know why the authors didn’t take it a step further and develop a systematic review? The article clearly shows there is research in the area warranting such an approach.

REPLY: We have considered a systematic review, however, we found this difficult to implement with nine articles in which the species varied and also the tests and testing conditions (factors such as age, administration duration and test parameters) employed and have thus maintained the current format.
Our concern was that we wouldn’t be able to make a good comparison of the articles then in accordance with the systematic review guidelines, which is why we had a more narrative approach in the body of the text. We did want to maintain transparency on why we have selected these exact articles and how we defined our search, which explains the systematic approach for the literature search.

Point 5: As the authors have used the more systematic approach it disrupts the overall flow of the manuscript. It might have been clearer and more in line with a traditional narrative review to incorporate the behavioral findings and the mechanisms in the same sections to allow the reader to tie the two areas together with greater ease?

REPLY: Thank you for your suggestion, we appreciate and understand your point. This was one of the considerations we had when we originally wrote the manuscript but we ultimately came to the current format. The reason why we separated the behavioural findings and the mechanisms is twofold, and done in an attempt to help the reader find the information they are looking for with ease:

  1. The first reason for separating the potential underlying mechanisms from the behavioural findings is because we consider those mechanisms to be potential contributing factors to the behavioural findings reported in the papers, but these mechanisms were not all directly investigated in those same papers. Therefore, we decided to report the biological mechanisms as potential contributors in our review text, but more research needs to be performed that investigates the direct link of these mechanisms with HMO functioning and downstream effects on cognition. This is why we made the distinction between the actual behavioural findings upon HMO administration reported in the 9 articles, and which mechanisms would potentially explain those findings.
  2. The second reason is that the behavioural findings are directly derived from the papers, and therefore, our third section in the review ‘Effects of HMOs on cognition in mammals’, can be seen as a  “results” section. The idea behind it is that a reader who just wants to have the main outline of the behavioural results can easily refer to our “results” section without having to search for the behavioural results in a larger chapter describing several related subjects. If the reader is then also interested in the possible underlying mechanisms, then those are neatly summarised in a separate section.

With these points in mind, we attempt to provide the information in a structured manner to make it easy to find for the reader, and we feel that the highly structured format we use enables this better than the traditional narrative format. For this reason we have decided to maintain the article’s current structure.

Point 6: Figure 1 and 2 gives a really nice visual overview of the types of behavioral tests and overall results. A really nice addition.

REPLY: Thank you for the compliment.

Point 7: Could the authors comment on the physiological relevance of the doses used in these studies. Would there be grams of each HMO in actual breastmilk or the human infant formula that contains HMOs?

REPLY: To our knowledge, the content of different HMOs in human milk ranges up from a few mg to several grams per L of milk, depending on a multitude of factors, such as the lactation stage of the mother; her diet, genetic variation, the specific HMO, and so on. The total HMOs in mature milk is usually in the range of 10-15 grams per liter (please see for example Austin et al., 2019: “Human Milk Oligosaccharides in the Milk of Mothers Delivering Term versus Preterm Infants”), with 2-‘FL levels being 3,72g/L in human breast milk, while 3-SL and 6-SL tend to be 6- 10 times lower than 2’-FL, and fall in the range of 150-250 and 600mg /L depending on lactation stage. 

As the 2’-FL studies reported in the review use mg/Kg bodyweight instead of g/L, this can be converted to mg/Kg bodyweight by taking the average infant weight and daily milk volume consumption into account. The average breast milk consumption volume per baby per day (in the first 6 months) is around 800 mL and the average body weight of an infant at 6 months old is roughly 8Kg, thus 100ml per Kg bodyweight. This translates to a 372mg/Kg BW 2’-FL intake, which is quite similar to the 350mg/Kg/BW reported in most of the 2’-FL studies. When considering 3’-SL and 6’SL content, this tends to be around 200- 250mg/L and 500-600mg/L in the first few weeks of lactation and then decreases to the 150-300mg/L range respectively over time. This is comparable to the dosage used in the sialyllactose studies as well.

Most studies cited in the review have thus considered the actual level of the specific HMOs to design their dosages and therefore the used doses are physiological relevant. In addition, while Wang et al. (2007) found a dose-response correlation upon supplementation of sialyllactose in a complex task, other studies with physiological appropriate dosages still yield results, so in light of this, the results are valid.

We have commented on its physiological relevance in the text when discussing the results (please see page 9, line 220-223) and cited relevant papers that provide a in depth comprehensive overview of HMO concentrations.

Point 8: Rodents are altricial and pigs are precocial, how does this affect brain development and might this account for some of the differences between the species in this line of research?

REPLY: Thank you for pointing this out. Yes, we do believe this might be a factor as the third trimester in neurodevelopment for humans coincides with either an in utero environment (piglets) or ex utero environment (rodents), thus the immediate environment, the nutrition content and the route of supplying this nutrition varies between species as well. Pigs are closer to human in terms of development during gestation and their brain morphology is also much more like human (e.g. gyrification) than rodents are.

We have expanded on this aspect further in the discussion of the results on page 9, lines 227-238 and made a mention of this in the conclusions on page 12 (line 373).

Point 9: In the discussion please make sure to identify which species the cited studies are referring to. This is especially important given that there are differences in the effects of HMO supplementation between piglets and rodents in the outlined studies.

REPLY: Thank you for your suggestion, we have now detailed the species in the discussion on pages 9 and 10, lines 214; 232-241; 246, 251, 277, 286.

Point 10: The authors point out the contribution of the gut and in particular the microbiome in mediating the effects of HMOs. Would the environments that each species are housed in influence the microbiome of each species in a way which may dictate the overall results?

REPLY: Thank you for drawing our attention to this. Indeed, we didn’t consider this aspect in our initial submission. Upon reviewing the literature we found the work of Lees and colleagues (2014), which indicates that housing conditions has marked influences on the microbiome. We have included this factor in the discussion, under section 4.1 “potential underlying mechanisms” on page 11, line 343-347.

Point 11: Each of the species studied produces litters, is there any information in relation to the sex of the animals who underwent the behavioral tests and might this influence the results? There is a growing body of result to suggest quite significant sex-specific outcomes in a range of different organ systems in response to supplementation (particularly when it is administered early in life) so a comment on sex-specificity and whether this may influence the ability of HMOs to produce an impact of neurocognition should be commented on. If you can find any information on whether human mothers produce differing amounts of specific HMOs in response to infant sex that would also be worth mentioning.

REPLY: These are two very interesting points and we thank you for bringing this up.

  1. Concerning potential gender differences in the studies:

The studies have reported the sex of the animals, but the vast majority (all except 2) of the studies used only males. Obelitz-Ryom et al. (2019) and Oliveros et al. (2018) have used a mixture of males and females in their sialyllactose studies, but in their analysis and results, no explicit separation based on sex has been made to our knowledge. It is therefore difficult to determine to what extent gender may have played a role in the results of these studies.

When both males and females were tested on the NORT at a later age (Oliveros et al. 2018) or males and females were tested at a young age in a more complex task (Obelitz-Ryom et al., 2019), the studies reported positive effects on behavioural outcomes upon sialyllactose supplementation. When comparing these two mixed gender studies with the other studies using the same animal model (e.g. Oliveros et al (2018) with Vazquez et al. 2016; and Obelitz-Ryom et al (2019) with Wang et al. (2007)), the results are very similar.

Therefore, based on these findings, the effects of HMO supplementation seem similar between both genders, but only two studies used a mixture of males and females (one piglet study, one rodent study), which makes it difficult to ascertain if and how gender plays a direct role or not in the context of cognitive effects of HMO administration. Nevertheless, we do agree that sex specificity in behavioural testing is an important factor to consider.

  1. Concerning gender specific HMO production: this is a very good question and it is true that we didn’t consider this factor in the original submission, thank you for calling our attention to this. According to the study from Azad et al (2018) (available at https://academic.oup.com/jn/article/148/11/1733/5105883), infant sex was largely unassociated with HMO composition in maternal milk. However, Moossavi et al. (2019) have reported that milk microbiota are dependent on infant sex, potentially due to an interaction with the gut microbiota which vary between males and females. As HMOs interact with both the milk and the gut microbiome (see also Ramani et al, 2018: https://www.nature.com/articles/s41467-018-07476-4), it is therefore possible that such variations in breastmilk microbiome composition could, lead to differential outcomes upon HMO supplementation.

We have included the first point as a brief discussion on page 10 (line 261 – 270). The second point has been added as a potential mechanism of interest in the microbiome section on page 11 (line 347-354) and included the appropriate references. Lastly, we also specified the gender of the animals in table 1, page 5-7.

Point 12: Line 284-285 – This sentence is confusing and should be reworded.

REPLY: Thank you for your suggestion, we have reworded the sentence to “The observation that HMOs are capable of enhancing cognition has initiated the search for a better mechanistic understanding of its functioning.” Please see page 12, line 356 – 357.

Reviewer 3 Report

The protective and long-lasting effects of Human Milk Oligosaccharides on cognition in mammals

The review aims to summarize the reported observations regarding the effects of HMOs on memory and cognition in mammals. The autors included of nine articles. The main findings illustrate that the administration of fucosylated and sialylated HMOs results in marked improved memory and learning outcomes. These results point towards the possibility that HMO administration, either in singular or combination forms, has long-lasting, beneficial effects on memory and cognition in mammals.

General

The paper is well written and interesting to read. The authors make a clear summary of the results obtained in the 9 papers included in the review. However, I see the following issues:

  • Line 30 “Studies have shown that exclusive breastfeeding is accompanied by a reduced risk for developing medical conditions during childhood such as gastrointestinal infections, asthma and, celiac disease [4]”   The cited work is not the most suitable. Bar et al mention 3 references (Eidelman AI, et al, 2012; Shelov SP, et al. 2009; and https://www.healthychildren.org/English/ages-stages/baby/breastfeeding/Pages/Why-Breastfeed.aspx.). But, more recent works should be cited. In addition, although retrospective studies seemed to indicate that breastfeeding reduced the risk of developing celiac disease, recent prospective studies have not been able to demonstrate this statement (Vriezinga SL, et al. Randomized feeding intervention in infants at high risk for celiac disease. N Engl J Med. 2014 Oct 2;371(14):1304-15) For this reason, the statement made in the introduction on reducing the risk of CD is controversial.

  • Table 1 could be horizontal, rather than vertical, so that the last column can be expanded, to read better. The sample size (n) / number of individuals studied from the 9 articles could be included.

  • Line 298 “Another important issue is that due to the large variability between the experimental design and methods used across studies, comparing the effects of different HMOs between studies is difficult. These limitations, ranging from age of testing, the tests themselves and the HMO components, call for a larger, unified study in which the effects of different HMOs on complex cognitive functioning are systematically compared, when administered both independently and as in conjunction.” The authors highlight the heterogeneity of the studies, which makes it difficult to compare them. Would the differences in the sample size of each also be important?

Author Response

Dear reviewer,

Thank you for your time and your feedback, we find your comments very helpful. We have replied to your comments in blue below and the changes in the manuscript are indicated in red. 

Response to Reviewer 3 Comments

General

The paper is well written and interesting to read. The authors make a clear summary of the results obtained in the 9 papers included in the review. However, I see the following issues:

Point 1: Line 30 “Studies have shown that exclusive breastfeeding is accompanied by a reduced risk for developing medical conditions during childhood such as gastrointestinal infections, asthma and, celiac disease [4]”   The cited work is not the most suitable. Bar et al mention 3 references (Eidelman AI, et al, 2012; Shelov SP, et al. 2009; and https://www.healthychildren.org/English/ages-stages/baby/breastfeeding/Pages/Why-Breastfeed.aspx.). But, more recent works should be cited. In addition, although retrospective studies seemed to indicate that breastfeeding reduced the risk of developing celiac disease, recent prospective studies have not been able to demonstrate this statement (Vriezinga SL, et al. Randomized feeding intervention in infants at high risk for celiac disease. N Engl J Med. 2014 Oct 2;371(14):1304-15) For this reason, the statement made in the introduction on reducing the risk of CD is controversial.

REPLY: Thank you for the compliment. This is a good point indeed, and we thank you for the literature reference.  Upon reviewing the latest literature regarding celiac disease, we have decided to omit the mentioning of celiac disease in the context of HMOs. In addition, we have included more recent references regarding the risk for developing medical conditions and breast milk in the introduction (please see page 1, line 32-35).

Point 2: Table 1 could be horizontal, rather than vertical, so that the last column can be expanded, to read better. The sample size (n) / number of individuals studied from the 9 articles could be included.

REPLY: Thank you for your suggestions. We have changed the table orientation to increase readability, and included the sample size of the animals used as well with the table, please see table 1, pages 5-7.

Point 3: Line 298 “Another important issue is that due to the large variability between the experimental design and methods used across studies, comparing the effects of different HMOs between studies is difficult. These limitations, ranging from age of testing, the tests themselves and the HMO components, call for a larger, unified study in which the effects of different HMOs on complex cognitive functioning are systematically compared, when administered both independently and as in conjunction.” The authors highlight the heterogeneity of the studies, which makes it difficult to compare them. Would the differences in the sample size of each also be important?

REPLY: The sample sizes for the rodent studies subset do not vary that much as n=8 - n= 12 is most commonly reported for rodents in these studies (with one exception using 28 mice per group), and the rodent studies produce rather similar results. When rodents were supplemented with HMOs, they perform better in complex tasks (operant tasks) in young adulthood (around 3-4 months old on average) and in simple tasks in mature adulthood (1 year of age).

In the case of the piglet studies, there is more variation in sample sizes within this subset, as sample size per experimental condition ranged from 12 to 20 animals. It is possible that, in the case of Obelitz-Ryom’s study (n= 12-20 piglets per group), the larger sample size could contribute to the sialyllactose results in the spatial T maze when tested at a young age, while Oliveros et al 2018 (n=10 rodents per group) were unable to produce such results on the Y maze and NORT at a young age. However, even with a larger sample size (n=17 piglets), Fleming and colleagues (2018) were unable to demonstrate any effects of sialyllactose supplementation on spatial memory measured by the NORT when they tested their piglets around the same age as Obelitz-Ryom and colleagues (2019) did.  When Fleming and colleagues changed the NORT parameters in their fucosyllactose studies in 2020 (n=12), they found a similar enhanced recognition memory with a sample size comparable to that of the fucosyllactose rodent studies (n=10), when the rodents were tested in mature adulthood. In addition, Wang et al. (20017) also found beneficial effects of sialic acid administration on the performance of a complex task in piglets (n=12 per group). Thus, even when piglet studies had comparable sample sizes to the rodent studies, the differences between species based on the age of testing remain prevalent.

Hence, while sample size could be a contributing factor to the discrepancies between the piglet and rodent studies, factors such as testing parameters, age and species differences seem to weigh more heavily. Nonetheless, in a unified study, the sample size could (and should) be kept the same to further decrease any possible confounding variation. In that sense, the sample size is definitely of importance.

We have included the sample sizes of the respective studies in Table 1 (page 5-7) and we also included a brief discussion on the possible contribution of sample size in the discussion section, line 271 - 276.

Round 2

Reviewer 2 Report

Thank you to the authors, my comments have been answered. Well done on an informative well written review. 

Author Response

Dear reviewer,

Thank you very much for the compliment.

With kind regards